# Maternal counseling for preterm deliveries, assessing an effective method of counseling: A randomized trial

**Shaaista Budhani[1], Mopelola Akintorin[1], Kenneth Soyemi[1], Louis Fogg[2], Mary Arlandson[3], Rajeev Kumar[1]\***

1 Department of Pediatrics, Division of neonatology, John H Stroger Hospital of Cook County, Chicago, Illinois, United States of America, 2 Division of Statistics, Rush University Medical Center, Chicago, Illinois, United States of America, 3 Department of Obstetrics and Gynecology, John H Stroger Hospital of Cook County, Chicago, Illinois, United States of America

\* rkumar3@cookcountyhhs.org (RK)

## Abstract

### Objective

To assess the acquired knowledge of mothers about prematurity outcomes when employing two distinct approaches to prenatal counseling among those experiencing preterm labor. A secondary aim was to assess the anxiety levels of trial subjects after the antenatal consultation.

### Study design

Ninety-two pregnant women admitted between 23 and 34 weeks of gestation with threatened preterm labor were randomized in to two groups to receive either verbal counseling (Group 1) or verbal counseling supplemented with written and pictorial information (Group 2). Mothers completed a validated anxiety inventory and demographic questionnaire before counseling and an anxiety inventory and knowledge questionnaire after the counseling. There was a mini-mum two-hour gap between the counseling and completion of the questionnaire.

### Results

Of the 92 women who completed the knowledge questionnaire, 45 (49%) were in Group 1 and 47(51%) in Group 2. Forty-three participants in group 1 and 45 participants in group 2 had their pre and post anxiety scores analyzed. There was a trend of increased recall rates in group 2 for short-term problems, long-term problems, intervention, and incidence rates, but it did not reach statistically significant level. There was an overall decrease in State Trait Anxiety Inventory (STAI) scores of participants after counseling (p = 0.002) but no statistically difference in change of STAI scores between the two groups (p = 0.981).

### Conclusion

Based on the results of our study, regardless of the method of counseling there was no difference in knowledge assessment and comprehension of information. However, there was an overall decrease in anxiety level of mothers following any type of counseling.

**Data availability statement:** Data has been uploaded to Mendley. Soyemi, Kenneth (2023), "MATERNAL COUNSELING FOR PRETERM DELIVERIES, ASSESSING AN EFFECTIVE METHOD OF COUNSELING: A RANDOMIZED TRIAL.", Mendeley Data, V1, doi: 10.17632/ry4v5b3f2h.1

**Funding:** The authors received no specific funding for this work.

**Competing interests:** The authors have declared that no competing interests exist.

## Trial registration

ClinicalTrials.gov NCT02707237

## Introduction

One of the most important aspects of medical management before a preterm delivery is counseling the parents about the probabilities of infant survival [1], short- and long-term morbidities. The American Academy of Pediatrics (AAP) and the American College of Obstetrics and Gynecology (ACOG) [2], have published guidelines for counseling of parents for births at the threshold of viability. But goals for counseling for more advanced gestational ages are unclear [2]. It has been suggested that written policies and procedures can facilitate consistent, effective, and timely counseling [3,4]. How to effectively provide complex information to women in premature labor is challenging [5]. Numerous studies have shown that supplementing verbal information enhances parental knowledge. Muthusamy et al. showed that providing written information along with verbal counseling increases parents' knowledge of long-term problems of prematurity [6]. Other methods to enhance parent's knowledge of survival and morbidities is to use visual aids such as pictures, graphics and short messages about resuscitation and complications associated with extreme preterm birth.[3]. Despite such benefits, it has been reported that two thirds of the hospitals in the US provide only verbal information [2].

With premature deliveries, there is little time between the counseling session and the delivery [7]. Parents are expected to comprehend and process the information provided to them during a compromised and often stressful state. The question then remains, how do we transfer and enhance the knowledge of women admitted to hospital in preterm and threatened preterm labor [8] in the limited time available to enable them to feel empowered to make informed decisions for their baby? To answer this question, we designed a study to explore and compare the effectiveness of verbal counseling vs verbal counseling supplemented with written and pictorial information. The study was conducted in an inner-city population and checked the recall rates of common problems of prematurity after counseling had been provided. We hypothesized that supplementing verbal counseling with written and pictorial information would improve the knowledge and understanding of complications and outcomes associated with prematurity among mother's facing a threatened preterm delivery. A secondary aim was to assess the anxiety levels of these women before and after the antenatal consultation [8].

## Methods

### Study design

The study was conducted at John H Stroger hospital of Cook County (Chicago, Illinois) between January 2016 and September 2017. The study ended after recruitment of the desired number of participants. The study protocol was approved by the Institutional Review Board of John H Stroger hospital of Cook County, Chicago, IL (15–226), ON 1/19/2016 and registered with the clinicaltrials.gov (NCT02707237) [9].

Participants were recruited when Obstetricians requested a consult for women admitted with threatened preterm delivery between 23 and 34 weeks. Consultation was provided by an Attending Neonatologist or a Neonatal fellow. Verbal consent to participate in the study was obtained by the physician providing the counseling and documented in the patient's chart. Participants were given an option to opt out from the study and still receive the same information. After enrollment participants were randomly assigned to two groups to receive counseling via verbal (Group 1) or verbal supplemented with written and pictorial information

(Group 2). Group randomization was achieved by using sealed opaque envelopes containing group numbers. State Trait Anxiety Inventory (STAI) was used to assess the anxiety in patients before and after the counseling (S1 Protocol).

Gestational age specific written material was used as a guide for verbal counseling ensuring standardization and consistency. Attending Neonatologists and fellows were trained to provide consistent information about delivery room resuscitation, common problems of prematurity, incidence rates of the short term and long-term outcomes.

### Inclusion criteria

Eligible participants included English and Spanish speaking women admitted to labor and delivery between 23- and 34-weeks' gestation in preterm labor or at risk of going into preterm labor and assessed by an obstetrician in need of counseling.

### Exclusion criteria

Pregnancies with known life-threatening defects considered as non-compatible with life, mothers previously admitted and counseled during the current pregnancy and mothers who delivered before counseling could be provided and non-English and non-Spanish speaking women.

## Materials

Counseling material contained gestational age specific information of delivery room management, common short-term, long-term complications of prematurity and incidence rates of survival. The written information used was at eighth grade reading level. These materials have been used in a previous study at a different institution and permission was obtained to use them in our study (6). Because of the variances in the education and socioeconomic status (S1 Appendix), written information was adapted for our patient population. We developed a pictorial booklet containing hand drawn and real patient pictures describing and depicting common problems of prematurity (S1 Fig.pdf). Material was made available in both English and Spanish (Education material was translated into Spanish from English by a native Spanish speaking Attending Physician). Hospital Spanish interpreter was used to provide translation when needed. We have piloted the information at our institution and presented the results of that study at a previous academic conference [10].

## Measures

### Knowledge questionnaire

A 32-item post counseling questionnaire was used to assess parental knowledge of short and long complications and outcomes of prematurity (S2 Appendix). A similar questionnaire was used in a previously published study and permission was obtained for use in this study [6]. The questionnaire was adapted for our patient population. Participants responded in a yes/no/don't know format to gestational age specific problems of prematurity and interventions. To assess the recall of incidence of select outcomes, participants wrote a number to indicate percent incidence of these outcomes. The Spanish translated version of the questionnaire was validated and used in a pilot study at our hospital [9]. Gestational age specific incidence rates for various complications of prematurity for our hospital were provided during the counseling, which were comparable to the "Vermont Oxford Data". The 'don't know' responses to items requiring yes/no/don't know responses were assessed as incorrect. A +/−20% range was allowed and accepted as correct in items requiring numerical responses.

### State trait anxiety inventory

The State Trait Anxiety Inventory (STAI) is validated in pregnant women and is used to detect change in the state of anxiety. The six-item Marteau and Bekker version of STAI which has been found to be reliable and valid was used with the inventory higher scores reflect higher states of anxiety.

### Patient characteristics

A knowledge questionnaire was designed to capture sociodemographic and religious characteristics, type of insurance, education levels, income, marital status, employment status, previous experience with premature babies, exposure to knowledge on prematurity prior to admission for threatened premature delivery. Income data was also collected for these families, and we used $20,000 median income as a cut off for poverty. For year 2016 Federal Register reference definition for poverty level was income below $16040 for a family of two and below $24300 for a family of four.

### Data analysis

Sample size was calculated with a desired power of 80% to detect a knowledge difference of 25% between the two groups, sample size was calculated as 44 participants in each group. Qualitative variables were expressed as frequencies and percentages. Quantitative variables were expressed as mean values (+/- standard deviation [SD]) if they followed a normal distribution or as median values (interquartile range [IQR]) if they did not. Association was tested with $X^2$ test or Fischer's exact test when necessary. The Shapiro-Wilk test was used to determine the normality of the continuous variables (age, hours, gravida, pre and post counseling STAI test scores).

Differences between two means were tested with the independent samples t-test, when normality conditions permitted. When normality conditions were not met, Mann-Whitney U test or Wilcoxon rank sum test was used to test differences in central tendencies. The repeated measure analysis of variance (ANOVA) with the between-group factor inclusion was used to determine if there was a statistical difference in the mean pre and post STAI tests for both intervention groups. Before conducting the ANOVA test, whether or not the data fits the assumptions of using this approach (normality, sphericity and equality of variance of measures among groups) were evaluated and passed.

## Results

Of the 92 women who completed the knowledge questionnaire, 45 (49%) were in Group 1 (verbal counseling only) and 47 (51%) in Group 2 (verbal counseling supplemented with written and pictorial information) [Fig 1]. Two participants in Group 1 and one participant in Group 2 did not complete the post counseling STAI questionnaire and were not analyzed.

Baseline characteristics of the participants were similar across both groups. Both groups were very similar in terms of maternal age, gestational age, race and ethnicity, economic status, and maternal education level (Table 1). The overall median (IQR) time between pre and post questionnaires was 5.5 hours (10.75).

There was no statistically significant difference in overall recall rates between the two groups for short term problems (such as respiratory distress, brain bleed, retinopathy of prematurity, feeding problems). The same was true for long term problems (chronic lung disease, cerebral palsy, behavioral problems, blindness), interventions needed (chest massage, oxygen, ventilator need), and incidence rates of long-term outcomes (such as survival, brain bleed, chronic lung disease and retinopathy of prematurity) (Table 2). Though

Women eligible for threatened preterm labor, n=130

Not enrolled in the study:
-Declined, n=3
-Romanian/Russian language speaking, n=2
-Life threatening anomalies,n=2
-<23 weeks gestation,n=1
-Consult not requested by Obstetrician, n=12
-Not approached for consent to participate, n=4
-Delivery before study completion, n=13

Enrolled in Study n=93

Mom discharged before posttest completed, n=1

Study completed, n=92

Group 1, n =45

Group 2, n=47

**Fig 1. Participant enrollment.**

participants remembered use of umbilical lines, feeding problems and blood transfusion better in Group 2 ($p < 0.05$).

The repeated sample analysis of variance that included the between-group effect was performed to determine if there was a statistically significant difference in the mean STAI scores before and after patients received one of the two types of counselling. The test

**Table 1. Baseline characteristics of participants.**

| Variable | Group 1 n = 45 | Group 2 n = 47 | P value |
|---|---|---|---|
| Gestational age(wks), mean ± SD | 29 ± 5 | 30 ± 3 | 0.94 |
| Maternal age (yr), mean ± SD | 28 ± 7 | 28 ± 6 | 0.62 |
| Maternal race n(%) | | | |
| Black | 28 (62) | 24 (51) | 0.29 |
| White | 15 (33) | 23 (49) | 0.19 |
| Other | 2 (4) | 0 (0) | 0.47 |
| Married n (%) | 11 (24) | 13 (30) | 0.41 |
| Education n (%) | | | |
| High School | 26 (57.8) | 31 (66) | 0.42 |
| College | 19 (42.2) | 16 (34) | 0.42 |
| Working n (%) | 14 (31) | 18 (38) | 0.48 |
| Income n (%) | | | |
| ≤$20,000/yr. | 31 (72) | 35 (79) | 0.56 |
| >$20,000/yr. | 12 (28) | 9 (21) | 0.40 |
| Previous premature delivery n (%) | 11 (24) | 17 (36) | 0.23 |
| Magnesium sulphate n (%) | 24 (53) | 20 (43) | 0.31 |
| Father of the baby involved n (%) | 39 (87) | 44 (94) | 0.44 |

p value < 0.05

revealed that while the change in the average anxiety scores decreased from before the training to after the training for both study groups, there was no statistically significant difference in mean difference of the STAI scores between the two groups ($p = 0.639$). Fig 2 below shows the pattern of changes in the mean score of the two groups before and after the counseling sessions. There was an overall decrease in STAI scores of participants after counseling (p=<0.001) but no statistical difference in change of STAI scores between the two groups (p = 0.98) (Table 3).

## Discussion

Our study sought to compared two different methods of antenatal counseling in an inner-city population of pregnant females with threatened preterm delivery and it showed no difference in recall rates of common problems of prematurity and incidence rates of outcomes, between the two groups. Although there was an overall improvement trend in the percentage of correct responses in Group 2 (68.5%) compared to Group 1 (61.8%), this was not statistically significant. It is possible that women experiencing threatened preterm labor are most concerned about their unborn baby, but they also have concerns regarding their family, their own health, financial stressors, and transportation issues [11], thus the actual medical information may not be as important to mothers [7] as the need for hope, reassurance and identifying the parents' future roles [11–13].

Muthusamy et al, demonstrated a statistically significant decrease in STAI scores among subjects who received detailed written information compared with the control group who received only verbal information and found no increase in psychological burden of the mother who received more detailed information [6]. Our study also showed an overall decrease in anxiety scores after counseling, although anxiety scores decreased in similar fashion in both the groups. The Muthusamy et al study also showed improvement in recall of long-term

**Table 2. Outcomes and interventions by group comparison.**

| Short Term Problems | | | |
| --- | --- | --- | --- |
| Variable n (%) | Verbal<br>n = 45 | VWP<br>n = 47 | P value |
| Respiratory distress | 39 (87) | 40 (85) | 0.8 |
| Apnea/bradycardia | 23 (50) | 25 (53) | 0.6 |
| Brain bleed | 33 (73) | 40 (85) | 0.2 |
| Retinopathy of prematurity | 34 (76) | 37 (79) | 0.7 |
| Infection | 32 (71) | 33 (70) | 0.9 |
| Patent Ductus Arteriosus | 20 (44) | 24 (51) | 0.5 |
| Feeding problems* | 31 (69) | 40 (85) | 0.01 |
| Necrotizing Enterocolitis | 19 (43) | 24 (51) | 0.3 |
| Jaundice | 31 (69) | 35 (75) | 0.6 |
| **Long Term Problems** | | | |
| Variable n (%) | Verbal | VWP | P value |
| Chronic lung disease | 26 (58) | 27 (57) | 0.9 |
| Cerebral palsy | 23 (51) | 28 (60) | 0.4 |
| Behavioral problems | 23 (52) | 31 (66) | 0.1 |
| Blindness | 27 (61) | 32 (68) | 0.4 |
| Recurrent respiratory infections | 21 (47) | 19 (40) | 0.6 |
| Kernicterus | 18 (41) | 21 (45) | 0.7 |
| Hearing loss | 22 (50) | 29 (62) | 0.2 |
| **Interventions** | | | |
| Variable n (%) | Verbal | VWP | P value |
| Chest massage | 21 (47) | 31 (65) | 0.06 |
| Oxygen | 23 (51) | 28 (60) | 0.4 |
| Ventilator need | 39 (87) | 42 (89) | 0.9 |
| Surfactant | 34 (75) | 37 (79) | 0.7 |
| Umbilical lines * | 26 (58) | 38 (81) | 0.01 |
| Feeding tube | 36 (80) | 38 (81) | 0.8 |
| Medicine for PDA treatment | 30 (66) | 31 (66) | 0.9 |
| Phototherapy | 34 (76) | 39 (83) | 0.4 |
| Antibiotic use | 35 (78) | 42 (89) | 0.1 |
| Blood transfusion | 35 (77) | 36 (77) | 0.9 |
| Surgery other than circumcision | 28 (62) | 31 (66) | 0.7 |
| **Recall of Incidence rates** | | | |
| Variable n (%) | Verbal | VWP | P value |
| Survival | 37 (81) | 35 (74) | 0.4 |
| Brain bleed | 16 (35) | 24 (51) | 0.1 |
| Chronic lung disease | 17 (37) | 19 (41) | 0.8 |
| Retinopathy of Prematurity | 20 (44) | 28 (59) | 0.1 |
| Blood transfusion * | 22 (48) | 33 (71) | 0.04 |

outcomes and incidence rates of mortality and morbidities of prematurity in the intervention group. In our study we did not see such difference, this could be explained by the higher education level and private insurance status of the participants (indicating higher socioeconomic status) in above study, as compared to our patient population which had mostly low education level and carried public insurance or no insurance at all.

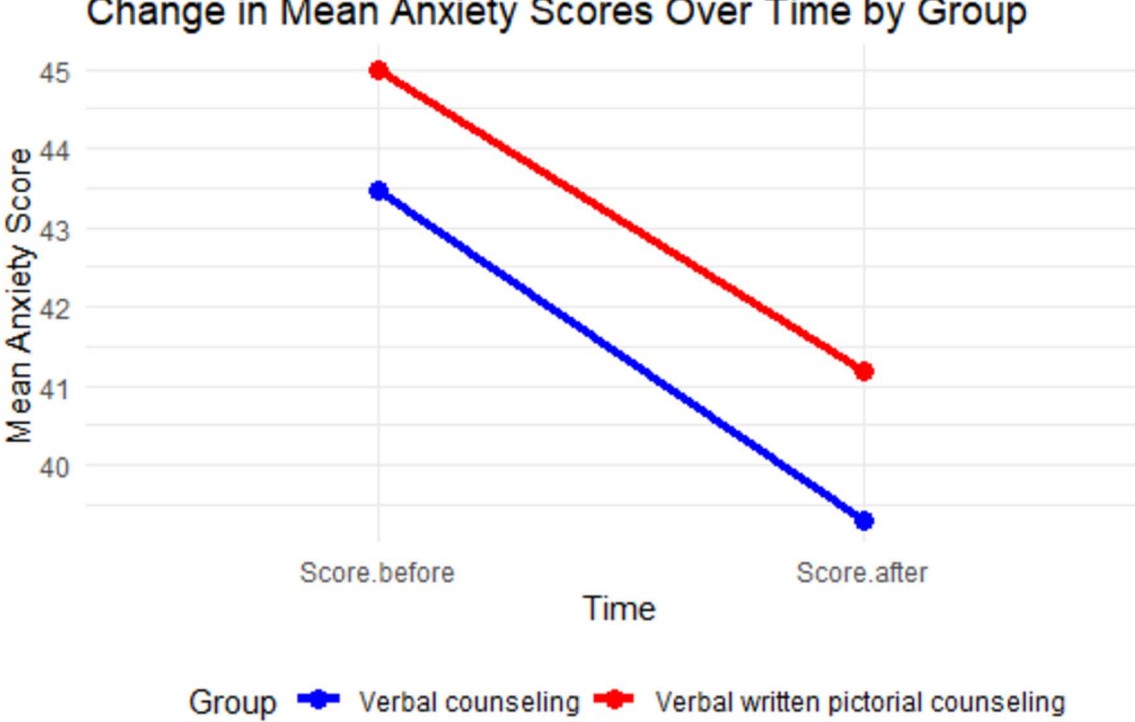

**Fig 2. Change in Mean Anxiety Scores Over time by group.**

**Table 3. Change in STAI scores after counseling.**

|  | Pre counseling STAI | Post counseling STAI |
|---|---|---|
| Verbal (n = 43), mean ± SD | 43.3 ± 13.6 | 39.3 ± 13.0 |
| VWP (n = 46), mean ± SD | 45.1 ± 13.4 | 41.1 ± 12.5 |
| Overall change,mean ± SD* | 44.3 ± 13.4 | 40.2 ± 12.7 |
| Income < $20,000 | 43.6 ± 12.9 | 40.1 ± 12.9 |
| Income > $20,000 | 44.9 ± 12.2 | 40.0 ± 12.4 |
| HS graduate | 43.8 ± 14.4 | 40.2 ± 12.7 |
| College Graduate | 44.8 ± 11.4 | 40.4 ± 12.9 |
| Self-Pay (No Insurance) | 54.5 ± 21.0 | 47.8 ± 15.4 |
| Insured | 44.0 ± 13.1 | 40.0 ± 15.4 |
| Education Combined | 44.2 ± 13.3 | 40.2 ± 12.8 |
| Income Combined | 43.9 ± 12.7 | 40.0 ± 12.7 |
| Insurance Combined | 44. 1 ± 13.5 | 40.1 ± 12.9 |

*p value < 0.05

In an earlier study including a sample of mothers who received a diagnosis of fetal malformations, it was demonstrated that maternal anxiety levels after delivery were inversely associated with the number of prenatal consultations [14,15]. However, since most mothers receive counseling during their admission for preterm or threatened preterm delivery, it may not be possible to meet with the mothers more than the initial encounter. Ninety-eight percent of our participants had only one encounter with the neonatologist prior to delivery.

Kaplan et al suggested that socioeconomic position exerts a pervasive influence on many health outcomes across the lifespan [16] including poor cognitive performance. Previous research has found that individuals with poor executive functioning (i.e., short term memory, working memory, verbal fluency, auditory attention) exhibit greater deficits in verbal and visual memory function [16–18]. The results of our study showed that those with higher incomes had a better overall recall rate although this effect was not seen between the two groups. Within the two groups combined 78.5% of our study participants had income levels of <= $20,000/year influencing the recall rates of our study population.

Studies identified in this field of counseling have had participants with high school education or higher [6] or required mothers to recall their experiences with antenatal counseling to develop decision aids [19]. Our study had 62% of participants with high school education or less and were in preterm labor or threatened preterm labor with a risk for imminent delivery. It is possible that in this population, counseling at the time of presentation in labor may not be efficient. It has been suggested that during routine prenatal care, risk factors for preterm delivery be identified and these mothers get counseling in a non-urgent setting allowing them time to process information.

This study had its limitations; 98% of the counseling was done by neonatal fellows at different levels of training. It is possible that variations in discussions may reflect the level of training. This study did not analyze parental responses based on the physician doing the counseling. Although our power calculation indicated that the study had the sample size required to assess the difference between the two groups- a larger sample may have elicited results that were statistically significant.

## Conclusion

Based on our study results, additional information above verbal counseling did not alter the knowledge assessment or comprehension of information, but there was an overall decrease in anxiety level of mothers after any type of counseling. Further studies able to utilize larger sample sizes, are needed to develop a counseling method that can effectively and efficiently transfer information to mothers in preterm labor and reduce anxiety at the same time.

## Supporting information

**S1 Protocol.** Study Protocol.
(TIF)

**S1 Appendix.** Sample counseling material.
(ZIP)

**S1 Fig.pdf.** Sample picture booklet.
(ZIP)

**S2 Appendix.** Sample questionnaire.
(ZIP)

## Author contributions

**Conceptualization:** Shaaista Budhani, Mopelola Akintorin, Mary Arlandson, Rajeev Kumar.

**Data curation:** Shaaista Budhani.

**Formal analysis:** Kenneth Soyemi, Louis Fogg.

**Investigation:** Mopelola Akintorin, Rajeev Kumar.

**Methodology:** Shaaista Budhani, Mopelola Akintorin, Mary Arlandson, Rajeev Kumar.

**Project administration:** Shaaista Budhani, Mopelola Akintorin, Mary Arlandson, Rajeev Kumar.

**Supervision:** Mopelola Akintorin, Mary Arlandson, Rajeev Kumar.

**Writing – original draft:** Shaaista Budhani, Mopelola Akintorin, Rajeev Kumar.

**Writing – review & editing:** Shaaista Budhani, Mopelola Akintorin, Kenneth Soyemi, Rajeev Kumar.

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
