## [Decision Letter · Decision Letter 0]

2 May 2023

PONE-D-22-35479MATERNAL COUNSELING FOR PRETERM DELIVERIES, ASSESSING AN EFFECTIVE METHOD OF COUNSELING: A RANDOMIZED TRIAL.PLOS ONE

Dear Dr. Kumar,

Thank you for submitting your manuscript to PLOS ONE. After careful consideration, we feel that it has merit but does not fully meet PLOS ONE’s publication criteria as it currently stands. Therefore, we invite you to submit a revised version of the manuscript that addresses the points raised during the review process.

We look forward to receiving your revised manuscript.

Kind regards,

Dumbani Lyspin Kayira, MBBS, MSc, FCPaed(SA)

Guest Editor

PLOS ONE

Journal Requirements:

2. In the ethics statement in the Methods, you have specified that verbal consent was obtained. Please provide additional details regarding how this consent was documented and witnessed, and state whether this was approved by the IRB

- https://doi.org/10.1016/j.jpeds.2016.08.006

- https://doi.org/10.1542/peds.2015-2336

- http://weirdthings.com/t8oxgd0/interpretation-of-wilcoxon-signed-rank-test

In your revision ensure you cite all your sources (including your own works), and quote or rephrase any duplicated text outside the methods section. Further consideration is dependent on these concerns being addressed.

4. Please include your ethics statement in the Methods section of your manuscript. In the Methods section of your revised manuscript, please include the full name of the institutional review board or ethics committee that approved the protocol, the approval or permit number that was issued, and the date that approval was granted.

5. Please upload the trial protocol and the ethics documents you submitted as separate files.

7. Please include a separate caption for each figure in your manuscript.

8. Please upload a copy of Figure 2, to which you refer in your text on page 7. If the figure is no longer to be included as part of the submission please remove all reference to it within the text.

Additional Editor Comments (if provided):

Abstract

Major comment: The results section within the abstract seems to focus on the secondary outcome. It does not present actual results for the primary outcome. Only a conclusion is presented saying there was no significant difference in recall rates between the two groups (which means the two counseling methods had similar effectiveness). Please present in a very concise manner the actual rates or measures of effect and/or p-values on which this conclusion is based. The results section should share the actual results on which the conclusion is based, as the abstract is stand-alone.

Introduction

Minor comment: As you present the purpose of the study at the end of the introduction, please very briefly touch on specifically assessing “recall rates” for your primary outcome. That is the main element you evaluated.

Minor comment: You have presented the purpose as a question. Please end with a question mark for punctuation.

Methods

Minor comment: Please provide a reference for the validated knowledge questionnaire which you mention was used in a previously published study (reference to that previous study)

Minor comment: Please provide a reference for the Spanish translation of the same knowledge questionnaire which you mention you validated in a pilot study at your institution (reference to this validation pilot study for the Spanish version)

Results

Major comment: You present secondary outcome (anxiety score) before the primary outcome, and in more detail. The primary outcome (recall rates) is only found as one sentence (more of a conclusion) in the last paragraph of the results section; there is no table or any number provided on the primary outcome. Yet as the primary outcome even your sample size calculation was based on detecting a knowledge difference between the two groups. Please swap around. Start with the primary outcome (recall rates) and provide adequate results on recall rates across the two groups for short term problems, long term problems, intervention, and incidence rates (provide numbers or even a table, and not just one concluding sentence in the last paragraph)

Minor comment: I am not sure what Figure 2 refers to. Please double check if there should be a reference to figure 2.

Reviewers' comments:

Reviewer's Responses to Questions

**Comments to the Author**

1. Is the manuscript technically sound, and do the data support the conclusions?

Reviewer #1: Yes

2. Has the statistical analysis been performed appropriately and rigorously? 

Reviewer #1: I Don't Know

3. Have the authors made all data underlying the findings in their manuscript fully available?

Reviewer #1: Yes

4. Is the manuscript presented in an intelligible fashion and written in standard English?

Reviewer #1: Yes

5. Review Comments to the Author

Reviewer #1: The paper reports on a small study which compares the effectiveness of verbal counselling versus verbal plus supplemental material for women threatened with preterm delivery. They found no significant differences. The study was reasonably well carried out but is not of much general interest to the broader medical/scientific community.

Other points.

There was no assessment of the literacy abilities of the individual in the two groups, and whether that affected their understanding of the written materials.

The STAI inventory (p6). is the short version of the STAI used based on the State rather than the Trait scale?

6. PLOS authors have the option to publish the peer review history of their article (what does this mean? ). If published, this will include your full peer review and any attached files.

**Do you want your identity to be public for this peer review?** For information about this choice, including consent withdrawal, please see our Privacy Policy .

Reviewer #1: No

---

## [Author Response · Author response to Decision Letter 1]

13 Oct 2023

MATERNAL COUNSELING FOR PRETERM DELIVERIES, ASSESSING AN EFFECTIVE METHOD OF COUNSELING: A RANDOMIZED TRIAL.

Dear Editors and Reviewers,

Thank you for your time and consideration in reviewing our manuscript. We appreciate your feedback. All comments have been addressed. Please see below for our responses.

Comments:

Response: Manuscript has been updated to meet PLOS ONE’s style as required.

2. In the ethics statement in the Methods, you have specified that verbal consent was obtained. Please provide additional details regarding how this consent was documented and witnessed, and state whether this was approved by the IRB

Response: Verbal consent explanation has been updated in the manuscript.

- https://doi.org/10.1016/j.jpeds.2016.08.006

- https://doi.org/10.1542/peds.2015-2336

- http://weirdthings.com/t8oxgd0/interpretation-of-wilcoxon-signed-rank-test

Response: Plagiarism has been addressed.

In your revision ensure you cite all your sources (including your own works), and quote or rephrase any duplicated text outside the methods section. Further consideration is dependent on these concerns being addressed.

Response: Duplicated text has been edited all the work has been cited (9,10).

4. Please include your ethics statement in the Methods section of your manuscript. In the Methods section of your revised manuscript, please include the full name of the institutional review board or ethics committee that approved the protocol, the approval or permit number that was issued, and the date that approval was granted.

Response: Ethics statement has been added. IRB name has been added.

5. Please upload the trial protocol and the ethics documents you submitted as separate files.

Response: Protocol and Ethics documents have been submitted as separate files.

Response: Data has been uploaded to Mendley.

Soyemi, Kenneth (2023), “MATERNAL COUNSELING FOR PRETERM DELIVERIES, ASSESSING AN EFFECTIVE METHOD OF COUNSELING: A RANDOMIZED TRIAL.”, Mendeley Data, V1, doi: 10.17632/ry4v5b3f2h.1

7. Please include a separate caption for each figure in your manuscript.

Response: Each figure has its own caption now. Figure has been moved to supplemental information.

8. Please upload a copy of Figure 2, to which you refer in your text on page 7. If the figure is no longer to be included as part of the submission please remove all reference to it within the text.

Response: Figure 2 reference has been removed.

Response: Captions for supporting documents have been added at the end of manuscript.

S1 Protocol: Study Protocol.

S1 Appendix: Sample counseling material

S1_Fig.pdf: Sample picture booklet

S2 Appendix: Sample questionnaire

Additional Editor Comments (if provided):

Abstract

Major comment: The results section within the abstract seems to focus on the secondary outcome. It does not present actual results for the primary outcome. Only a conclusion is presented saying there was no significant difference in recall rates between the two groups (which means the two counseling methods had similar effectiveness). Please present in a very concise manner the actual rates or measures of effect and/or p-values on which this conclusion is based. The results section should share the actual results on which the conclusion is based, as the abstract is stand-alone.

Response: Result section has been modified to reflect primary outcome.

Introduction

Minor comment: As you present the purpose of the study at the end of the introduction, please very briefly touch on specifically assessing “recall rates” for your primary outcome. That is the main element you evaluated.

Response: Introduction was modified to reflect “Recall rates” as primary outcome.

Minor comment: You have presented the purpose as a question. Please end with a question mark for punctuation.

Response: Punctuation has been added.

Methods

Minor comment: Please provide a reference for the validated knowledge questionnaire which you mention was used in a previously published study (reference to that previous study)

Minor comment: Please provide a reference for the Spanish translation of the same knowledge questionnaire which you mention you validated in a pilot study at your institution (reference to this validation pilot study for the Spanish version)

Response: Reference citation has been added for knowledge questionnaire on page. (6)

Response: Reference and citation added for Spanish questionnaire. (10)

Results

Major comment: You present secondary outcome (anxiety score) before the primary outcome, and in more detail. The primary outcome (recall rates) is only found as one sentence (more of a conclusion) in the last paragraph of the results section; there is no table or any number provided on the primary outcome. Yet as the primary outcome even your sample size calculation was based on detecting a knowledge difference between the two groups. Please swap around. Start with the primary outcome (recall rates) and provide adequate results on recall rates across the two groups for short term problems, long term problems, intervention, and incidence rates (provide numbers or even a table, and not just one concluding sentence in the last paragraph)

Response: Result section modified to reflect primary outcome and table (table 2) was added to reflect the results of primary with p values.

Minor comment: I am not sure what Figure 2 refers to. Please double check if there should be a reference to figure 2.

Response: Figure 2 refence was removed.

Reviewers' comments:

Reviewer's Responses to Questions

Comments to the Author

1. Is the manuscript technically sound, and do the data support the conclusions?

Reviewer #1: Yes

2. Has the statistical analysis been performed appropriately and rigorously?

Reviewer #1: I Don't Know

3. Have the authors made all data underlying the findings in their manuscript fully available?

Reviewer #1: Yes

4. Is the manuscript presented in an intelligible fashion and written in standard English?

Reviewer #1: Yes

5. Review Comments to the Author

Reviewer #1: The paper reports on a small study which compares the effectiveness of verbal counselling versus verbal plus supplemental material for women threatened with preterm delivery. They found no significant differences. The study was reasonably well carried out but is not of much general interest to the broader medical/scientific community.

Response: Counseling of parents facing a threatened preterm delivery is a big challenge for neonatologists, as anxiety levels are very high. Parents are very emotional at this time, as there life is about to change and they could be numb to information provided. This study explored the methods of delivering vast information in concise and easily understandable way to a innercity population. This study adds to pool of existing studies and raises a point of exploring methods that deliver information in more effective way.

Other points.

There was no assessment of the literacy abilities of the individual in the two groups, and whether that affected their understanding of the written materials.

Response: Parents education level taken as proxy of their literacy abilities.

The STAI inventory (p6). is the short version of the STAI used based on the State rather than the Trait scale?

Response: STAI inventory (p6) has a correlation coefficient of 0.9 with full version and is used in situations where full form is inappropriate similar to our study population

Marteau TM, Bekker H. The development of a six-item short-form of the state scale of the Spielberger State-Trait Anxiety Inventory (STAI). Br J Clin Psychol. 1992 Sep;31(3):301-6. doi: 10.1111/j.2044-8260.1992.tb00997.x. Erratum in: Br J Clin Psychol. 2020 Jun;59(2):276. PMID: 1393159.

6. PLOS authors have the option to publish the peer review history of their article (what does this mean?). If published, this will include your full peer review and any attached files.

Do you want your identity to be public for this peer review? For information about this choice, including consent withdrawal, please see our Privacy Policy.

Reviewer #1: No

6. Muthusamy AD, Leuthner S, Gaebler-Uhing C, Hoffmann RG, Li SH, Basir MA. Supplemental written information improves prenatal counseling: a randomized trial. Pediatrics. 2012;129(5):e1269-74. Epub 20120409. doi: 10.1542/peds.2011-1702. PubMed PMID: 22492766.

9. Akintorin M. Maternal Counseling for Preterm Deliveries, Assessing an Effective Method of Counseling Bethesda, MD: National Library of Medicine; 2016 [cited 2016]. Available from: https://clinicaltrials.gov/study/NCT02707237.

10. Rajeev Kumar MA, Christian Castillo, Medha Kamat, LouFogg,. Randomized control study comparing different methods of counseling in mothers at risk of imminent preterm delivery Pediatric Academic Societies (PAS); May 3rd-May 6th 2014; Vancouver , Canada2014.

---

## [Decision Letter · Decision Letter 1]

18 Dec 2023

PONE-D-22-35479R1MATERNAL COUNSELING FOR PRETERM DELIVERIES, ASSESSING AN EFFECTIVE METHOD OF COUNSELING: A RANDOMIZED TRIAL.PLOS ONE

Dear Dr. Kumar,

Thank you for submitting your manuscript to PLOS ONE. After careful consideration, we feel that it has merit but does not fully meet PLOS ONE’s publication criteria as it currently stands. Therefore, we invite you to submit a revised version of the manuscript that addresses the points raised during the review process.

Note that this round of feedback follows from a rigorous statistical review.  Please address the statistical recommendations provided. We hope that this will strengthen the science in this work that you have done.

We look forward to receiving your revised manuscript.

Kind regards,

Dr. Dumbani Kayira, MBBS, MSc, FCPaed(SA)

Guest Editor

PLOS ONE

Reviewers' comments:

Reviewer's Responses to Questions

**Comments to the Author**

1. If the authors have adequately addressed your comments raised in a previous round of review and you feel that this manuscript is now acceptable for publication, you may indicate that here to bypass the “Comments to the Author” section, enter your conflict of interest statement in the “Confidential to Editor” section, and submit your "Accept" recommendation.

Reviewer #2: (No Response)

2. Is the manuscript technically sound, and do the data support the conclusions?

Reviewer #2: No

3. Has the statistical analysis been performed appropriately and rigorously? 

Reviewer #2: No

4. Have the authors made all data underlying the findings in their manuscript fully available?

Reviewer #2: Yes

5. Is the manuscript presented in an intelligible fashion and written in standard English?

Reviewer #2: No

6. Review Comments to the Author

Reviewer #2: A two-arm randomized clinical trial was conducted which aimed to compare the effectiveness of verbal counseling to verbal counseling plus written and pictorial information on preterm deliveries. The conclusions are unclear.

Major revision:

Test the interaction of group by time rather than repeatedly testing differences in time and by group. If the interaction effect is significant, provide an interpretation of the results, but do not test main effects because the tests for main effects are uninteresting in light of significant interactions. If interaction effects are non-significant, drop the interaction effects from the model and test the main effects. Determining which results to present when testing interactions is often a multi-step process.

Minor revisions:

1- Study Design section: Indicate if block randomization was used. If so, state the block size.

2- Data Analysis section:

- State the alpha level and statistical testing method used in sample size calculation.

- Qualitative variables are typically expressed as frequencies and percentages.

- The independent t-test is written as “t-test”.

- The complete name of the Mann-Whitney test is the Mann-Whitney U test and it is used to compare differences in central tendencies rather than means.

3- The following sentence is unclear: “The p tests for the Shapiro-Wilk test for the continuous variables were p = 0.07 for Age; p = 0.00 for hours, gravida, and gestational age; p = 0.01 and 0.03 for pre and Post STAI scores meaning that these variables were not normally distributed.”

4- P-values never equal zero; express small p-values as < 0.001.

5- The Wilcoxon Signed Rank test is a nonparametric test which compares central tendencies between two groups.

6- Add numeric p-values to Table 1.

7- Cite the statistical software used for the analysis.

8- Thoroughly proofread the manuscript. Provide as much clarity as possible. A few noted grammatical mistakes follow.

- Study Design section: Be consistent with formats for the months of January and September.

- A few sentences are missing a period at the end.

9- To assist in the review process, add line numbering to the document.

7. PLOS authors have the option to publish the peer review history of their article (what does this mean? ). If published, this will include your full peer review and any attached files.

**Do you want your identity to be public for this peer review?** For information about this choice, including consent withdrawal, please see our Privacy Policy .

Reviewer #2: No

---

## [Author Response · Author response to Decision Letter 2]

1 Mar 2024

Dear Editor,

I have edited the manuscript as requested for grammatical errors, as well as statistical changes as requested. I hope I have answered all the questions to your satisfaction.

Kind Regards

Rajeev Kumar

---

## [Decision Letter · Decision Letter 2]

11 Mar 2024

PONE-D-22-35479R2MATERNAL COUNSELING FOR PRETERM DELIVERIES, ASSESSING AN EFFECTIVE METHOD OF COUNSELING: A RANDOMIZED TRIAL.PLOS ONE

Dear Dr. Kumar,

Thank you for submitting your manuscript to PLOS ONE. After careful consideration, we feel that it has merit but does not fully meet PLOS ONE’s publication criteria as it currently stands. Therefore, we invite you to submit a revised version of the manuscript that addresses the points raised during the review process.

We look forward to receiving your revised manuscript.

Kind regards,

Dr. Dumbani Kayira. MBBS, MSc, FCPaed(SA)

Guest Editor

PLOS ONE

Journal Requirements:

Reviewers' comments:

Reviewer's Responses to Questions

**Comments to the Author**

1. If the authors have adequately addressed your comments raised in a previous round of review and you feel that this manuscript is now acceptable for publication, you may indicate that here to bypass the “Comments to the Author” section, enter your conflict of interest statement in the “Confidential to Editor” section, and submit your "Accept" recommendation.

Reviewer #2: (No Response)

2. Is the manuscript technically sound, and do the data support the conclusions?

Reviewer #2: (No Response)

3. Has the statistical analysis been performed appropriately and rigorously? 

Reviewer #2: (No Response)

4. Have the authors made all data underlying the findings in their manuscript fully available?

Reviewer #2: (No Response)

5. Is the manuscript presented in an intelligible fashion and written in standard English?

Reviewer #2: (No Response)

6. Review Comments to the Author

Reviewer #2: Minor revisions:

1- Line 156: Wilcoxon is misspelled.

2- Line 156: Consider a test of the interaction effect of group by time instead of repeatedly using paired t-tests. "The paired t- test or the Wilcoxson rank signed test was used to determine if there was a statistical difference in the mean pre and post STAI tests for both intervention groups."

3- Line 180: The Wilcoxon signed rank test is not used to compare differences in means. It compares differences in central tendencies.

7. PLOS authors have the option to publish the peer review history of their article (what does this mean? ). If published, this will include your full peer review and any attached files.

**Do you want your identity to be public for this peer review?** For information about this choice, including consent withdrawal, please see our Privacy Policy .

Reviewer #2: No

---

## [Author Response · Author response to Decision Letter 3]

27 Mar 2024

Dear Editor,

We have revised the statistics with the help of professional statistician and all the minor changes have been addressed.

Kind Regards

Rajeev Kumar

---

## [Editor Report · Decision Letter 3]

11 Apr 2024

MATERNAL COUNSELING FOR PRETERM DELIVERIES, ASSESSING AN EFFECTIVE METHOD OF COUNSELING: A RANDOMIZED TRIAL.

PONE-D-22-35479R3

Dear Dr. Kumar,

We’re pleased to inform you that your manuscript has been judged scientifically suitable for publication and will be formally accepted for publication once it meets all outstanding technical requirements.

Kind regards,

Dr. Dumbani Kayira, MBBS, MSc, FC Paed(SA)

Guest Editor

PLOS ONE
---

## [Editor Report · Acceptance letter]

PONE-D-22-35479R1

MATERNAL COUNSELING FOR PRETERM DELIVERIES, ASSESSING AN EFFECTIVE METHOD OF COUNSELING: A RANDOMIZED TRIAL.

Dear Dr. Kumar:

I'm pleased to inform you that your manuscript has been deemed suitable for publication in PLOS ONE. Congratulations! Your manuscript is now with our production department.

Kind regards,

on behalf of

Dr. Dumbani Lyspin Kayira

Guest Editor

PLOS ONE